# Detection and In-Depth Analysis of Causes of Delay in Construction Projects: Synergy between Machine Learning and Expert Knowledge

**Marija Z. Ivanović \*, Đorđe Nedeljković, Zoran Stojadinović, Dejan Marinković, Nenad Ivanišević and Nevena Simić**

Department of Construction Project Management, Faculty of Civil Engineering, University of Belgrade, 11000 Belgrade, Serbia
* Correspondence: mapetrovic@grf.bg.ac.rs

**Abstract:** Due to numerous reasons, construction projects often fail to achieve the planned duration. Detecting causes of delays (CoD) is the first step in eliminating or mitigating potential delays in future projects. The goal of research is unbiased CoD detection at a single project level, with the ultimate goal to discover the root causes of delay. The existing approach is based on expert knowledge which is used to create CoD lists for projects in general or groups of similar projects. When applied to a single project, it is burdened with bias, as shown on a case project returning low Spearman Rank correlation values. This research introduces a Delay Root causes Extraction and Analysis Model—DREAM. The proposed model combines expert knowledge, machine learning techniques, and Minutes of Meetings (MoM) as an unutilized extensive source of information. In the first phase, DREAM outputs a CoD list based on occurring frequency in MoM with satisfactory recall values, significantly reducing expert-induced subjectivism. In the second phase, enabled by MoM dates, DREAM adds another dimension to delay analysis—temporal CoD distribution. By analyzing corresponding informative charts, experts can understand the nature of delays and discover the root CoD, allowing intelligent decision making on future projects.

**Keywords:** causes of delay; machine learning; transformers; bias; Spearman rank correlation; construction projects

## 1. Introduction

Construction projects are unique, dynamic, and complex. They also have feedback processes, non-linear relationships, and many project stakeholders [1], which produce a massive corpus of data. These features, as well as complex relationships between stakeholders, can affect the project's success [2]. In addition to traditional goals (iron triangle—time, cost, and quality) [3], recent construction projects should also satisfy sustainable development aspects. Achieving sustainable processes in project management is still a challenge. However, failing to achieve conventional goals may influence other project success parameters. According to most authors, time is apostrophized as one of the most critical project goals [4]. Despite its importance, the history of the construction industry testifies to many projects completed with significant time overruns [5]. Accordingly, it is necessary to re-evaluate existing approaches in this field of construction management.

Construction projects are especially prone to extreme events, i.e., black swans [6], which can significantly disrupt the project flow. Many authors consider that collecting information and experience on success from previous projects can contribute to managing future projects [7,8]. Therefore, a detailed analysis of disturbances from completed projects is an essential source of information and can help decision makers with future projects. Public projects are specifically sensitive to major disruptions such as pandemics or financial crises, significantly impacting their funding [9] and risking their viability. However, the

conventional management of this information becomes laborious and error-prone. Artificial Intelligence (AI) can help to transform accumulated data into useful knowledge [10], leading to intelligent decision making in construction projects.

Identifying the causes of delays is the first and essential step in analyzing project delays. Although more than 50 years have passed since the first research on this topic [11], delays are still present in construction projects. A literature review in Section 2 shows that, throughout various studies, similar delay-related issues are often "recycled" [12]. Most studies are based on empirical approaches to collecting and analyzing data on the causes of delays, mainly relying on questionnaires or interviews with experts. Their content is descriptive, with causes of delay identified and ranked according to the experience and knowledge of the experts. Such an approach is a valuable source of knowledge, but it is not in the context of issue-solving [13]. In addition, reliance on experts' judgment can lead to subjectivism and bias, raising the question of the reliability of the results [8,14].

Previous research on the topic and the characteristics of construction projects imply that it is imperative to analyze the causes of delay to address the question of *what exactly went wrong* at the individual project level. Such an in-depth analysis of the causes and circumstances leading to disruptions can be used to mitigate delays and enable intelligent decision making in future projects. This claim is supported by [8], which states that it is helpful to distinguish between causes and root causes of delay, leading to long-term benefits in construction management.

This article focuses on developing a model for identifying and analyzing the *causes of delay* (CoD) that will reduce expert bias and subjectivism and identify the *root causes*. The authors propose a new approach based on CoD detection from textual sources—DREAM (*Delay Root causes Extraction and Analysis Model*). Apart from CoD detection at the project level, the model will be able to group CoD into *separate physical entities* (from now on Entities) in the project, i.e., with the types of structures. The model introduces a new attribute for describing the CoD—*time*, which enables a more detailed cause analysis throughout the project timeline. In-depth CoD analysis by the project Entities and through the phases of implementation contributes to the identification of root CoD. *Text mining* algorithms based on *transformer models* were used in this research as a tool for extracting CoD-related knowledge from massive *unstructured textual documentation* in a construction project. Synergetic fusion of *machine-learning-provided information* and *users' expert knowledge* aims to unlock more options regarding CoD analysis, such as detecting the events leading to them. Based on that, DREAM produces strategic knowledge, contributing to intelligent decision making and sustainable management in future construction projects. Additionally, owing to Minutes of Meetings statements, unbiased results were obtained with contributions by all project participants. The value of these results is the reduction in disputes on projects.

This research has two main goals:

1.  To propose a framework for using text mining for detection and in-depth analysis of causes of delays in construction projects and to test the abilities of such an approach;
2.  To apply the proposed approach at a single project level, a still challenging research topic.

Both goals are valuable contributions to the knowledge of investigating delay causes in construction projects, aiming to overcome the biased judgment of experts and enhance practicality and usefulness.

## 2. Research Background

The exhaustive research of previous scientific studies comprises two phases: the first phase covered articles related to the causes of delays in construction projects; the second phase analyzed articles about the application of AI, especially text mining, in construction project management.

### 2.1. Causes of Delay

This paper emphasizes unbiased CoD identification and root CoD analysis but also covers academic research on different construction projects and this topic's general aspects. Their importance is reflected in the lists of identified CoD in construction projects and represents this research's starting point.

During the CoD-related literature analysis, the focus was on the methodology of CoD detection and the resulting CoD lists. Rankings and conclusions related to the identified causes at the individual article level were out of the scope of this research.

Most studies used a questionnaire to identify CoD [4,5,15–22]. The contribution of these studies is related to the CoD in different types of construction projects, various geographical regions, and from the point of view of other stakeholders. For example, Mahamid I. et al. investigated CoD in road construction projects in the West Bank in Palestine [19], Frimpong Y. et al. presented causes of delay and cost overruns in Ghana groundwater projects [20], Assaf S. et al. investigated CoD for different types of large construction projects in Saudi Arabia [15], Fallahnejad M. studied CoD in gas pipeline projects in Iran [17], Kaming P. et al. presented factors causing delays in high-rise projects in Indonesia [21], etc. Most studies focused on CoD in the *construction phase*. However, some studies analyzed them in the *planning and design phase* [23].

Exploring the perception of CoD by different stakeholders is essential. A study [16] conducted a questionnaire, which included project owners, architects, structural engineers, service engineers, project managers, contract administrators, design managers, and construction managers. One of the most commonly used metrics for examining the degree of agreement between the views of different stakeholders is the Spearman rank correlation non-parametric test. Table 1 presents an overview of the Spearman rank correlation values for selected studies. Other matters were obtained in further studies, concluding that no specific trend can be observed. Participants' perception of CoD may depend on the circumstances and the degree of delay in the project. Banobi E. et al. analyzed CoD from the perspective of the contractor and the owner in power construction projects in Tanzania [22]. The study showed that disagreement between the stakeholders increased with the delay in the project.

**Table 1.** Minimum and Maximum values of Spearman Rank correlation for pairs of project stakeholders regarding causes of delay.

| Authors | Year | Type of Construction Projects | Geography Region | Spearman Rank Correlation |
|---|---|---|---|---|
| Assaf S. et al. [15] | 2006 | Construction projects | Saudi Arabia | 0.568, 0.724 |
| Sambasivan M. et al. [5] | 2007 | Construction projects | Malaysia | 0.772, 0.896 |
| Le-Hoai L. et al. [24] | 2008 | Construction projects | Vietnam | 0.572, 0.776 |
| Abd El-Razek M. E. et al. [25] | 2008 | Construction projects | Egypt | 0.47, 0.69 |
| Enshassi A. et al. [26] | 2009 | Construction projects | Developing country | 0.421, 0.595 |
| Aziz R. et al. [27] | 2013 | Road | Egypt | 0.666, 0.838 |
| Fallahnejad M. [17] | 2013 | Gas pipeline projects | Iran | 0.710, 0.846 |
| Atibu Seboru M. [28] | 2015 | Road | Kenya | 0.64 |
| Bajjou M. et al. [29] | 2018 | Construction projects | Morocco | 0.939, 0.983 |
| Rachid Z. et al. [30] | 2019 | Construction projects | Algeria | 0.58, 0.64 |

On the other hand, a significantly smaller number of researchers use qualitative research methods, such as interviews. The main reason is a more demanding research method, primarily regarding the required resources, especially time. A study [12] prioritizes qualitative research, especially since it strives to understand the "real-world" issues, which is extremely important regarding the causes of project delays. The same research includes a qualitative CoD study conducted in 41 interviews with experienced practitioners in the U.K. The interviews were audio recorded and transcribed, and analyzed in NVivo software. The conclusion was that the qualitative approach provided a deeper CoD analysis.

### 2.2. Text Mining

The application of text mining is not widespread in construction projects [31]. Text mining in the construction domain has three primary purposes [32]: information retrieval [33,34], document classification [35], and knowledge discovery [36–39]. Due to the volume of unstructured textual data generated throughout the project lifecycle, there is significant potential for the application of text mining in construction project management. It is especially beneficial to combine the results obtained using text mining techniques with the results of various structured data analyses so that the stakeholders get better project insight [38].

Text mining techniques were used in [40] for the classification of texts related to construction site accident claims. The authors concluded that the main benefit of the proposed approach was the reduction in processing time, i.e., the saving of human resources in the documentation classification procedure. A model for extracting meaningful patterns from the construction project contracts and correspondence using BIM was developed in [31]. The authors emphasized the importance of the visual presentation of textual documentation for finding hidden patterns and trends. An invitation to Bid documents was used in [41] to develop a contract risk analysis tool for contractors, using artificial intelligence and text-mining techniques. The study provided a system of supporting project risk analysis and decision making by extracting and evaluating risk automatically.

The potential of the application of text mining techniques on the Minutes of Meetings (MoM) for the extraction of relevant construction project information is also recognized. Key phrases extracted from the MoM were analyzed according to their association with different stakeholders and distribution in time [42]. Similarly, authors in [43] used data extracted from the MoM texts to develop a predictive early warning methodology for project failure. Since their introduction in [44], transformer models have been widely used for text analysis as a type of deep learning language Machine Learning model. Transformer-based approaches are also used in research related to the construction industry. In [45], a general pre-trained BERT model was used to retrieve relevant paragraphs containing infrastructure damage information. The fine-tuned model was capable of identifying the answers to the questions related to the areas damaged by tropical cyclones and earthquakes. Authors used the GPT-2 model to reconcile the differences between look-ahead planning tasks and master-schedule activities based on the task descriptions [46]. In [47], the authors utilized a transformer-based model to recognize communication entities occurring in ICT patents in construction, resulting in improved performance over the baseline deep learning model. Similarly, [48] fine-tuned the BERT model for detecting near misses in safety reports, displaying improved performance over five other text classification methods.

There is no doubt that previous research identified the causes of delays from various aspects. Most of them, however, aim to draw general conclusions about the causes of delays for a defined market or/and type of construction project. On the other hand, different views of different stakeholders are justified, but they make it difficult to reach conclusions, especially at the level of an individual project. Based on the review of existing studies, an approach that can detect CoD at the single project level while reducing expert bias and subjectivity is needed. This research aims to improve the identification and analysis of the causes of project delays through the synergy of machine learning and expert knowledge. While project delay manifestation can be observed and measured more easily, identification of circumstances leading to them in the first place is a more complex task. The process of detecting and understanding these events, and the causality between them, can benefit from Machine-Learning algorithms. It is especially applicable to algorithms that use unstructured data and allow the extraction of relevant information from the text.

## 3. Research Structure

This section explains the research structure in detail. Figure 1 shows the steps in conducting the research (two approaches—upper and lower section) and corresponding levels of observation (presented in the left, middle, and right areas).

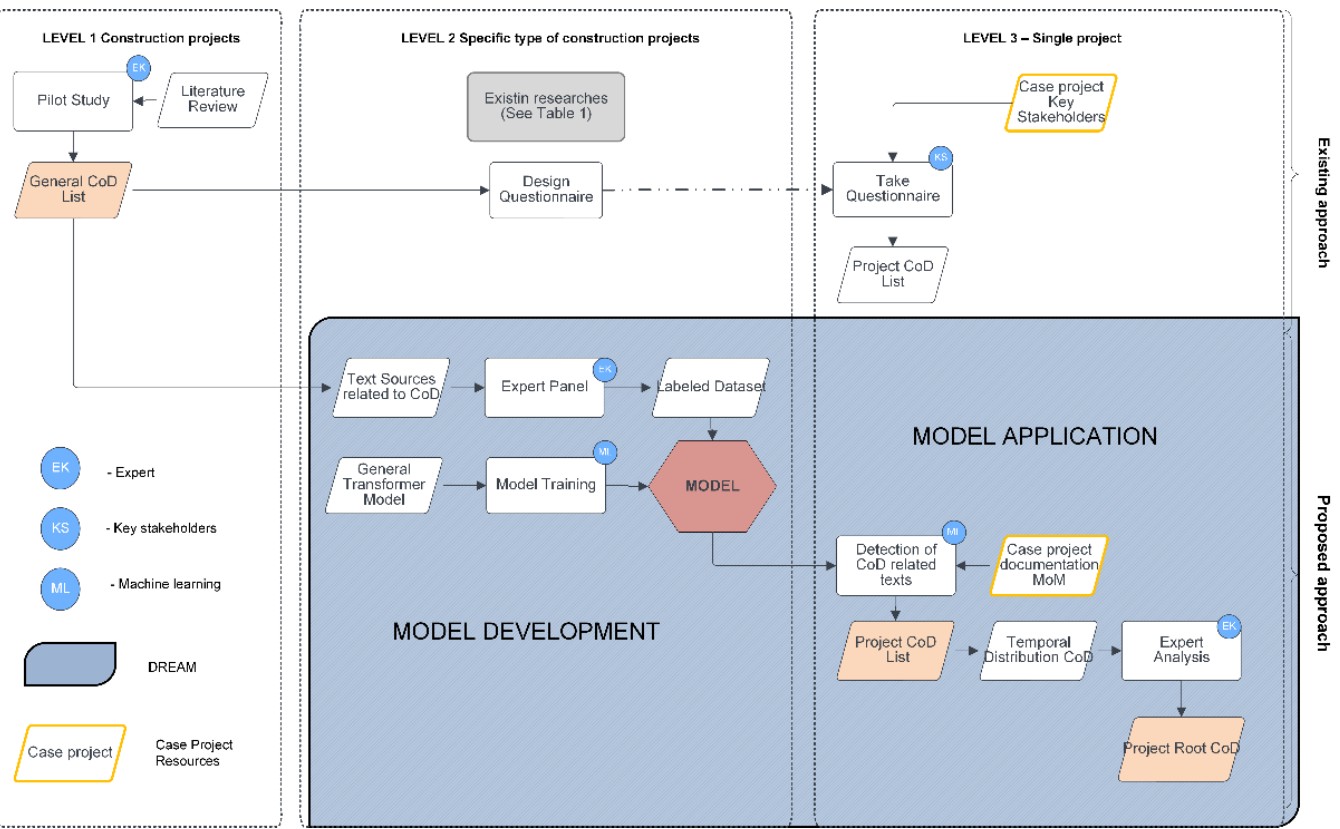

**Figure 1.** Research structure and key outputs: General Causes of Delay (CoD) List for the existing approach; CoD detection model, Project CoD, and Project Root CoD for the proposed approach (DREAM).

The upper part of Figure 1 shows the existing empirical approach to CoD detection:

- The upper-left part of Figure 1 presents a survey-based method for generating a generalized CoD list for construction projects (used by both approaches). Creating such a CoD list was the first step in this research.
- The upper-middle part of Figure 1 relates to applying empirical surveys to different project types or regions. The corresponding section presents a literature review of existing studies.
- The upper-right part of Figure 1 refers to survey applications at the single project level. The survey conducted in this research through a questionnaire shows different perspectives of three groups of key stakeholders (employer, contractor and engineer) about CoD.

The key outputs of this part of the research are the General CoD list and the single project CoD list based on the survey.

The bottom part of Figure 1 shows the proposed DREAM approach. The DREAM uses the General CoD List as a reference point for the design model. The middle part of the proposed approach presents the design of the DREAM. The main inputs for the model design are a labeled dataset and the general transformer model (DistilBERT), which result in multiple "text to CoD" classification models. This phase also evaluates the performance of the proposed approach in detecting the CoD-related text segments for a road infrastructure project. The resulting CoD-related text segments are organized in a list, allowing experts to analyze CoD temporal distribution to detect the root causes of delay. The right part is a single project-level DREAM application. The key outputs of this part are the DREAM, single project level CoD, and Root CoD based on the DREAM.

An example of the latter is given below to clarify the difference between causes and *root causes*. For instance, *rework due to errors or poor quality during construction* is a cause

recognized in most studies and expected to impact the project implementation deadline. However, there is a basis for an in-depth analysis to determine the circumstances leading to this cause. Errors or poor quality of works can reasonably be attributed to the contractor's performance. Therefore, the root cause of delay, in this example, can be attributed to the *poor performance of the contractor*. Additional information, such as temporal distribution and unbiased peer review, is needed to confirm this claim.

DREAM introduces a segmented model, according to the project type and its separate entities, capable of detecting CoD at the single project level. An essential part of DREAM is the segmentation of CoD by Entities—different physical parts or structures of a project. This research proposes three Entity types for road infrastructure projects:

- Tunnel—underground works;
- Route—civil engineering works;
- Bridge—structure.

Different structures aggregate specific delay causes, expected to be recognized by the proposed model. For the single project level, the research uses a case project—a highway in Serbia. The case project consists of two sections (A = 28 km and B = 37 km) with a total length of 65 km, contracted by Red Fidic. The contractor was an international company, while the employer and the engineer were domestic public companies. At the time of data collection, the road was open for traffic. The total project delay was 42% of the contracted duration.

## 4. Expert-Based Approach—A Survey

This section presents the results of the existing approach, based on expert opinion, for detecting the causes of delays. The first step is a literature review and a pilot study, resulting in a General CoD List that applies to all construction projects. The second step refers to the survey, which includes a questionnaire design and a discussion of the results of the case project. The goal is to analyze the applicability of surveys to a single project level.

### 4.1. Generating the General CoD List

The General CoD List was generated in three steps. The first step was a *literature review* related to CoD in construction projects (Section 2.1). A relevant literature search resulted in 74 academic articles (from 1997 to 2021) that contained CoD lists. The union of CoD lists from all articles resulted in a comprehensive list of 157 CoD. The second step was *merging duplicates*, resulting in a list of 94 CoD. As the number of CoD in the reviewed research ranged from 34 [49] to 83 [50], it was necessary to reduce the number of CoD so that the list is adequate for understanding and analysis.

The third step refers to the pilot study (a standard method for generating a CoD list used in similar studies) [4,5], aiming to identify previously unrecognized causes of delay, filter the proposed CoD list, and improve the structure of the list. The pilot study included three academic professors with more than 20 years of project management experience with whom individual interviews were conducted. In addition to the CoD recognized in previous studies, the pilot study also contributed by identifying three new causes of delay (2.2, 2.3, and 6.4), Appendix A. After making changes suggested in the pilot study, a General CoD List was formed, consisting of 52 CoD grouped into 8 categories (Appendix A). In addition to defining elements of the CoD list, the pilot study outcome also included suggestions for the structure of the survey itself, with a focus on making it more efficient. The most important recommendation was to have the optimal number of answer options for the sake of clarity.

### 4.2. Questionnaire Design

The questionnaire consists of two parts: (1) general information about respondents and (2) General CoD List presented in Appendix A.

The experts were asked to rate the degree of importance of CoD using a four-point Likert scale (Table 2). Following the recommendations given in the literature [51], the option "*I do not know*" was added as a response option to reduce subjectivism. The first version of the questionnaire was a standard five-point Likert scale (1—no or very low, 2—low, 3—average, 4—high, and 5—very high) [16], but the feedback from experts suggested they had difficulty distinguishing between 3–4 and 4–5. Therefore, the authors chose to reduce the Likert scale to a four-point version [15] to obtain more precise results. Any reduction in the number of possible answers improves correlation outcomes and vice versa. If the original scale were adopted, the correlation results would only be worse. Since the aim is to show that biased experts generate low correlation, the Likert scale reduction is justified.

**Table 2.** Four-point Likert scale for measuring CoD importance for project delays.

| 0 | 1 | 2 | 3 |
| --- | --- | --- | --- |
| No impact or very low impact | Low impact | Median impact | High Impact |

Spearman's rank correlation non-parametric test is used to measure and compare the relationship between the views of different stakeholders, for two each party, while ignoring the third party. This non-parametric test is calculated according to Equation (1):

$$r_s = 1 - \frac{6 \sum d^2}{n^3 - n} \tag{1}$$

$r_s$—Spearman rank correlation; $d$—the difference between ranks for each CoD; and $n$—the number of pairs of rank.

The value of the correlation coefficient varies in the interval from +1 to −1, where +1 implies absolute agreement while −1 is absolute disagreement. The limit values (−1 and +1) imply high correlation (negative or positive) whereas values close to zero indicate low or no correlation [15]. The obtained values will be used for discussing the reliability of the questionnaire results.

### 4.3. Findings and Results

This section discusses the degree of agreement between stakeholders regarding CoD importance and the applicability of expert surveys on the individual project level, while particular survey results remain outside the scope of this article.

The answers from stakeholder pairs defined three groups of results (Table 3). Every stakeholder had more than 20 years of experience in their respective role. The focus is on the degree of agreement between different stakeholders. The values of Spearman rank correlation show that there is a low correlation between every two groups of parties. The highest degree of correlation (0.36) is between owner and consultant, while the lowest is between contractor and consultant (0.24). These results are expected according to the position of the consultant on projects, whose role is to represent the owner. The low correlation coefficient is illustrated by the fact that only one cause of delay (3.2, see Appendix A) was detected by all three stakeholders. In addition, the same trend was observed with all three participants—shifting responsibility for the delay to the other contracting party. For example, according to the contractor, the dominant CoD were issues related to *the design* and *financing of the project*. On the other hand, the owner deemed that the most significant delays were related to the *contractor's productivity* and *poor project management*. The consultant presumed that the causes of delays relate to the *lack of manpower, equipment breakdowns, and obsolete equipment*. Based on these findings, it is clear that the participant's opinions about the CoD depend on their role in the project.

**Table 3.** Value of Spearman rank correlation for different stakeholder pairs.

| Parties | Spearman Rank Correlation *rs* |
| --- | --- |
| Contractor—consultant | 0.26 |
| Contractor—owner | 0.34 |
| Owner—consultant | 0.36 |

The survey results indicate a lack of consensus among the key stakeholders in the case project. The causes of the disagreement may be related to the characteristics of the stakeholders. Namely, the contractor was a company with no previous experience in the Serbian construction industry. On the other hand, the owner and the engineer were public companies, so external pressures were likely not in favor of the project. The circumstances of the above-mentioned scenario could have led to inadequate communication between stakeholders. An additional reason leading to lower values may be a limited number of survey respondents (three in this case), which raises a question about the general applicability of the questionnaire for the single project level.

Key stakeholders' opinions are a significant source of knowledge for identifying the project CoD. However, their role in the project can produce a bias that can affect the reliability of the results. This part of the research concludes that there is a need for improving the approach to CoD detection and analysis that will be resistant to subjectivism and bias and suitable for the single project level.

**5. Proposed Approach—Delay Root Causes Extraction and Analysis Model**

This paper presents a new approach to CoD detection and analysis of a construction project, aiming to reduce the effects of subjectivism and bias by applying machine learning techniques to unstructured textual data.

This section presents an overview of the proposed Delay Root causes Extraction and Analysis Model (DREAM). The First part explains the components required for model building, followed by DREAM experimental scenarios, validation procedure, and results. Finally, temporal distribution analysis is introduced as a unique feature of the proposed approach, which enables the detection and analysis of the root causes of delay.

*5.1. DREAM Components*

This section presents methodological requirements and workflow for building the proposed DREAM approach. The model utilizes expert knowledge and understanding of the causes of delay for a particular construction project type and machine learning techniques for text mining. The key components required to build the model are:

- MoM as documentation used for model training and validation;
- Expert labeling procedure for CoD-related text segments;
- Transformer models used for CoD detection.

5.1.1. Minutes of Meetings—Selected Documentation Type for DREAM

Out of all data on construction projects, around 80% are unstructured text documentation [31]. Alsubaey M. et al. (2015) argued that text data mining is vital for construction projects because there is a large amount of text data, such as meeting minutes, contracts, reports, or correspondence [43]. Apart from the construction phase, there is a significant amount of data before construction starts, such as contracts, project scope documents, specifications, or plans.

This research uses the case project minutes of meetings (MoM) for experiments (Figure 1). The potential of MoM was also recognized in a study [43], presented as project documentation providing early warnings of factors that led to project failure. It is expected that this type of written exchange of project information will point out disruptions that may threaten the project's goals. The significance of MoM is the connection between the disruptive event, the time of its occurrence, and the stakeholder reporting it. Unlike other

documents, MoM is created in an interactive environment by multiple stakeholders, making it difficult to manipulate its content. Since construction projects can last for several years and MoM is prepared weekly, a large volume of text data is inevitable. Expert interpretation of all MoM for a particular project would be complicated, time consuming, and prone to errors.

On the case project, meetings were held weekly, using a single MoM form throughout the project, with the following structure:

- General data (meeting number, time, and location);
- Participants present;
- List of statements discussed at the meeting, sorted by category (e.g., health and safety, quality of works).

The number of MoM per section and period is shown in Table 4. For continuing project topics, statements are repeated in MoM. In this research, *only statements with original content* were used for model training and validation, avoiding duplicating textual content.

**Table 4.** General MoM structure for the observed project.

| Section | No. of MoM | Period Covered | Original Statements |
|---|---|---|---|
| 1 | 64 | January 2018 to February 2020 | 1501 |
| 2 | 62 | February 2018 to February 2020 | 1411 |

### 5.1.2. Expert Labeling

The selection of MoM as documentation for model training was followed by expert labeling of the source text segments that could be linked to the particular CoD. For this phase of the research, an expert panel was organized, consisting of three distinguished practitioners in the field of construction management, specializing in Civil Engineering. They were also engaged in the pilot study and thoroughly familiar with the General CoD List from the first part of this research. One expert was a permanent consultant to the employer, the second was an occasional consultant to the contractor, and the third expert had no direct contact with the case project.

The panel carried out the protocol presented in Figure 2. The experts were required to label the MoM statements that referred to a particular CoD (Table 1). In addition, they tagged the corresponding project Entity (route, bridge, tunnel, or miscellaneous) for statements. The entity labeling was performed for all statements, regardless of whether they related to CoD or not.

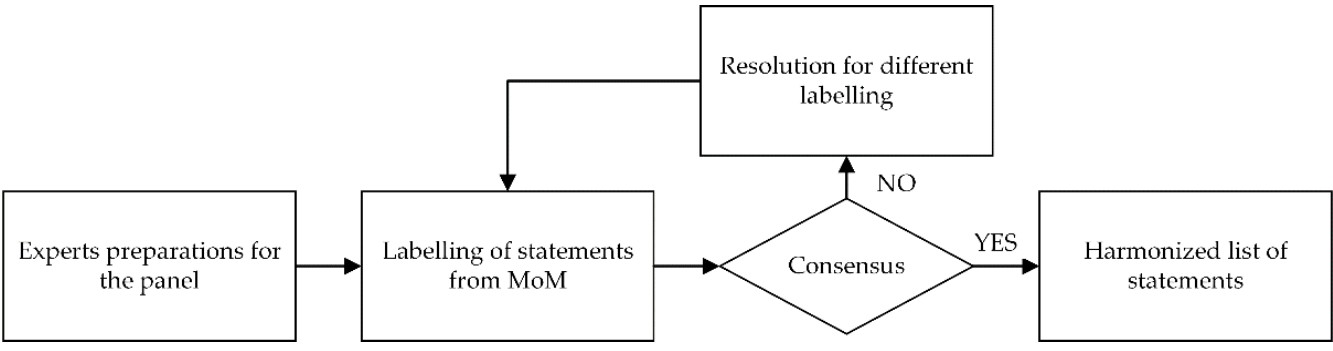

**Figure 2.** Procedure of experts' text labeling.

The criteria for reaching experts' consensus is if two or more experts performed labeling of a statement the same way. In the case of a non-consensus, a discussion followed to harmonize opinions. The expert panel lasted six months, reaching a consensus for labeled statements after three rounds. Table 5 shows an example of expert labeling.

**Table 5.** Examples of original statements labeled by experts.

| Cause of Delay | Element Type | Statement |
|---|---|---|
| 5.1 | Bridge | The contractor urgently corrected the deficiencies on the bridge at km 27 + 241. |
| - | Misc. | Expert supervision received the 3 most critical variations and evaluation is underway. The next meeting regarding the variations will be held on Wednesday. 18 February 2021. |
| 1.1 | Route | The cables are in the roadbed of the existing road IB and the employer will deliver a solution to the contractor in the second half of February. |

The purpose of expert labeling is to associate unstructured text in MoM with CoD, enabling the generation of a dataset consisting of MoM text segments linked with respective CoD. Such domain-specific datasets finetune the transformer model, adding an additional layer of semantic understanding on top of the existing general language model. The ultimate goal is to "teach" the model to associate CoD to corresponding, previously unseen, text segments generated in a construction project environment.

The result of the expert labeling is a total of 497 CoD-related statements, sorted into five CoD Groups and four Entities (Table 6).

**Table 6.** Frequency of labeled statements, by project Entities and CoD groups.

| Cause of Delay Group | Bridge | Route | Tunnel | Misc. |
|---|---|---|---|---|
| 1 | 11 | 99 | 8 | 4 |
| 3 | 6 | 69 | 18 | 6 |
| 4 | 0 | 14 | 5 | 0 |
| 5 | 16 | 119 | 13 | 16 |
| 8 | 0 | 76 | 9 | 8 |

5.1.3. Transformer-Based Models for Natural Language Processing

The proposed DREAM approach uses the existing general transformer language representation models, which are fine-tuned to reflect the context of the causes of delay in a construction project environment.

Transformer models are based on transfer learning methods, responsible for the breakthrough in Natural Learning Processing (NLP) and text mining in recent years. Transfer learning relies on using complex pre-trained models and their fine-tuning to provide high performance for various data mining tasks. Researchers from Google introduced transformer architecture [44]. The key differentiator from other Machine Learning approaches to language understanding, such as Recurrent neural networks and long short-term memory neural networks, was using a *self-attention* mechanism. It proved especially suitable for a deeper understanding of the text's longer sequences [44]. In their research, the authors state that "*self-attention, sometimes called intra-attention, is an attention mechanism relating different positions of a single sequence to compute a representation of the sequence*".

Transformer representation usually has an encoder–decoder architecture, with modules that contain feed-forward and attention layers. *BERT* (Bidirectional Encoder Representations from Transformers) language representation model is composed of transformer encoder blocks. The model was trained on Wikipedia (~2500 million words) and open-sourced in 2018.

Generally, language models read the input sequence in one direction: left to right or right to left. This kind of one-directional training works well when the aim is to predict/generate the next word. To have a more profound sense of language context, BERT uses bidirectional training, taking into account previous and next tokens simultaneously. To achieve this, BERT uses:

- Masked Language Modeling—masking of tokens in a sequence with a masking token and directing the model to fill that mask with an appropriate token. This allows the model to focus on right and left contexts (tokens on the right or left of the mask).
- Next Sentence Prediction—the model receives pairs of sentences as input and is trained to predict if the second sentence is the following sentence to the first or not.

As a result of the bidirectional training, the pre-trained BERT model can be fine-tuned with just one additional output layer to create state-of-the-art models for a wide range of text analysis tasks. BERT has 24 layers of transformer blocks with a hidden size of 1024 and 340 million parameters. This may challenge model training and utilization in the production environment. The authors of [52] introduced *Distil-BERT*, with 97% of BERT's performance and a 40% reduction in size while being 60% faster. The new model was created by leveraging knowledge distillation during the pre-training phase. Knowledge distillation is a compression technique in which a small model is trained to reproduce the behavior of a larger model (or an ensemble of models).

In this research, the pre-trained Distil-BERT model was fine-tuned with text segments reflecting the CoD context in the construction project environment. To address causes of delay with fewer text segments for training, the authors explored the capabilities of another transformer model—PEGASUS (Pre-training with Extracted Gap-sentences for Abstractive Summarization) [53]. PEGASUS uses an encoder–decoder model for sequence-to-sequence learning. In such a model, the encoder will first consider the context of the whole input text and encode the input text into a context vector, which is a numerical representation of the input text. This numerical representation will then be fed to the decoder, whose job is to decode the context vector to produce the abstractive summarization. Abstractive summarization is based on paraphrasing meaningful words from the input text resulting in synthetic output text examples with similar meaning and context to the input text. In this research, synthetic text statements related to CoD were generated by the PEGASUS model and added to the training set in one of the experiments to increase the number of training instances for less-represented CoD.

### 5.2. Experimental Scenarios, Validation Procedure, and Results

This section presents the performance of DREAM for different CoD detection tasks, respective experimental scenarios and setup, validation procedure, and results. The goals of the experiments were:

- Evaluation of the performance of the model for the classification of statements according to the level of individual CoD, CoD Group, and Project Entity;
- Determining the effect of applying the model segmentation process at the Project Entity level;
- Determining the effect of increasing the number of CoD-related statements through abstractive summarization.

### 5.2.1. Experiment Scenarios

Five experiment scenarios were defined and validated to achieve the previously determined goals. Figure 3 presents respective classification models, organized according to:

1. Target attribute for the classification:

    Project Entity;
    CoD Group;
    Individual CoD.

2. Method used for the model training:

    General: all labeled statements are used for training and testing;
    Segmented: labeled statements that refer to a particular project Entity are used for training and testing;
    Expanded: the number of labeled statements used for training is increased through abstractive summarization, based on the Pegasus model.

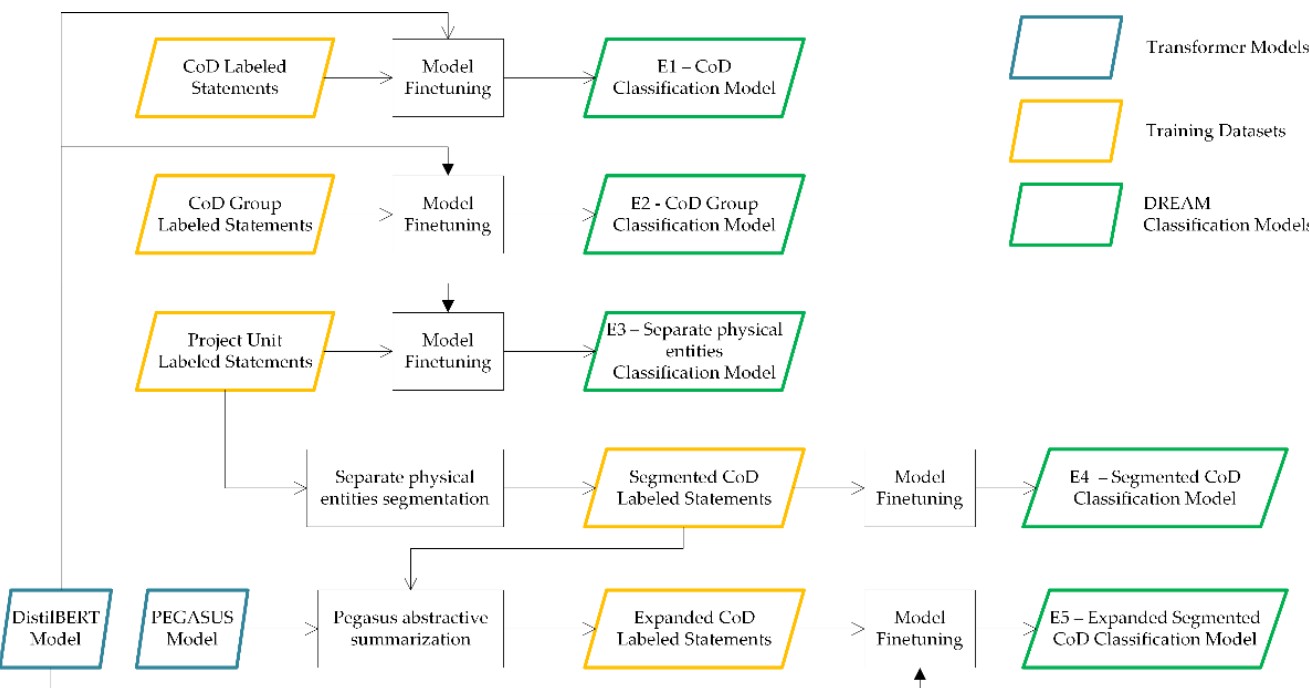

**Figure 3.** DREAM experiment scenarios. The first three experiments used general training and produced three separate classification models (E1—CoD level, E2—CoD Group level, and E3—Project Entity level). The fourth experiment (E4) was conducted with segmented training, where the training dataset was sliced according to a particular project Entity (route, bridge, and tunnel). The final experiment (E5) combined segmented and expanded training methods.

In all five experiments, the DistilBert transformer model was fine-tuned with the following hyperparameters:

- Learning rate algorithm—adaptive moment estimation 5e-5;
- Batch size—2 (maximum allowed by the GPU);
- Number of epochs—10;
- Max sequence length—512.

The experiments were performed on a server with an Intel© Core™ i3-10100F CPU @ 3.60 GHz × 4. NVIDIA Corporation GP107 (GeForce GTX 1050 Ti), with 4GB video memory GPU and 32 GB RAM.

### 5.2.2. Validation Procedure

The evaluation of the model was carried out through a 10-fold cross-validation procedure, using 90% of instances to train the model. The model was tested on the remaining 10% of instances. Test instances were chosen randomly but corresponded to class distribution at the level of the entire data set. This procedure was repeated ten times so that all labeled statements were used for training and testing in different folds. In each fold, the actual and predicted values for the testing instances were recorded and later combined into a list of actual and predicted values for all instances. This list was used for creating the confusion matrix to present and discuss the experiment results.

Since the MoM also contained statements that were not associated with any CoD, a special class was introduced for them in the classification model. These statements were labeled as the non-CoD (NC) class. In each cross-validation fold, the total of N statements was from the NC class, where N equals 90% of the total number of CoD statements. A ratio of 90% was chosen to model the representation of NC statements in real-life MoM adequately.

An MxM confusion matrix was used to evaluate the classification model's performance, where M is the number of classes for prediction. The matrix shows the actual values in

rows and predicted values in columns. The ideal scenario is that all predicted and actual values match, in which case the matrix has non-zero values only on the main diagonal. As the number of instances per individual class varies, the matrix is shown here in a normalized form (the number of elements in each class is 1.00, and a separate column with the total number of instances per class is presented). As rows represent the actual values of classes, the row sum is always 1.00. In contrast, the column sum can be different from 1.00, depending on the behavior of the classification model. A class with a sum greater than 1.00 is an *attractor* class (more instances than expected are assigned to it). In the opposite case, the class is a *repeller*, with fewer instances than expected.

5.2.3. Experimental Results for the General Models (E1, E2, and E3)

The E1 experiment classified statements according to individual CoD and covered the entire data set. Its results indicate better performance of more numerous CoD. The following specific trends can be observed in the resulting confusion matrix (Figure 4):

- More numerous CoD, in addition to better overall performance, were also attractor classes in most cases. The most probable cause was the fact that fewer common CoD did not have a sufficient number of examples to convey their underlying semantic structure to the model. As a consequence, most of them were associated with more common CoD;
- A significant number of CoD from group one were classified as CoD 1.5, while none were associated with their own class (values for the 1.1—1.4 on the main diagonal are 0). The CoD 1.5 classification occurred 61 times, which was the same as all others from Group 1 combined, resulting in 1.5 being the attractor class for the whole group. A similar trend can be observed for Group 3;
- Notably, the NC class absorbed a significant number of less common CoD. This was expected because the NC class was the most represented one in the model, while at the same time, it had the most general statements in terms of linguistic structure.

A more detailed expert interpretation of the results of E1, taking into account the nature and characteristics of the causes and respective groups, shows that:

- Although the model did not have many training instances for CoD 8.7 (unfavorable weather conditions), it performed best. This behavior is expected, primarily because of the linguistic nature and the way of reporting this cause through the MoM. Some erroneous classifications such as CoD 3.2 (low productivity and unqualified workforce) and 5.1 (rework due to errors or poor quality during construction) are also expected due to possible correlation to and mutual influence of these CoD on 8.7. Statements may have mentioned unfavorable weather conditions affecting the quality of the works and the productivity of the workforce at the construction site.
- CoD 1.5 (delays in the preparation or modification of design documentation during construction) had some false classifications in Group 8 (external). This can be explained by the relation of CoD 1.5 with external factors. For example, 8.2 (delay in obtaining permits and approvals from the competent authorities) may be associated with 1.5 in the case of delays in property–legal relations at that particular location. It can be concluded that, due to the nature of the project, the primary cause of the delay in some situations may be 8.2 and not 1.5. Interpretation of the confusion matrix can reveal similar trends, which can be used for the analysis of the correlation between causes.
- There is a trend of false classification between groups 3 (resource) and 5 (contractor). A possible explanation can be traced to the logic of forming groups 3 and 5 and the type of contract used in the case project. In this case, it may be advisable to merge these two groups into one because most of the causes from group 3 relate to the contractor and the performance of the contractor's workforce.

| | | predicted | | | | | | | | | | | | | | | | | | | | total |
|---|---|---|---|---|---|---|---|---|---|---|---|---|---|---|---|---|---|---|---|---|---|---|
| | | 1.1 | 1.2 | 1.3 | 1.4 | 1.5 | 3.2 | 3.4 | 3.5 | 4.3 | 4.4 | 5.1 | 5.3 | 5.4 | 5.6 | 5.7 | 8.1 | 8.2 | 8.6 | 8.7 | NC | |
| actual | 1.1 | 0 | 0 | 0 | 0 | 0.28 | 0.11 | 0 | 0 | 0 | 0 | 0.11 | 0 | 0.11 | 0 | 0 | 0.06 | 0 | 0 | 0 | 0.33 | 18 |
| | 1.2 | 0 | 0 | 0 | 0 | 0.50 | 0.17 | 0 | 0 | 0 | 0 | 0 | 0 | 0.17 | 0 | 0 | 0 | 0 | 0 | 0 | 0.17 | 12 |
| | 1.3 | 0 | 0 | 0 | 0 | 0.39 | 0.06 | 0 | 0 | 0.11 | 0 | 0 | 0 | 0 | 0 | 0 | 0.11 | 0 | 0 | 0 | 0.33 | 18 |
| | 1.4 | 0 | 0 | 0 | 0 | 0.46 | 0.08 | 0 | 0 | 0 | 0 | 0.08 | 0 | 0 | 0 | 0 | 0 | 0 | 0 | 0 | 0.38 | 13 |
| | 1.5 | 0 | 0 | 0 | 0 | 0.69 | 0.02 | 0 | 0 | 0 | 0 | 0.02 | 0.02 | 0.07 | 0 | 0 | 0 | 0 | 0 | 0 | 0.2 | 61 |
| | 3.2 | 0 | 0 | 0 | 0 | 0 | 0.73 | 0 | 0 | 0 | 0 | 0.04 | 0.02 | 0 | 0.05 | 0 | 0 | 0 | 0 | 0.01 | 0.14 | 83 |
| | 3.4 | 0 | 0 | 0 | 0 | 0.14 | 0.43 | 0 | 0 | 0 | 0 | 0.14 | 0 | 0 | 0 | 0 | 0.14 | 0 | 0 | 0 | 0.14 | 7 |
| | 3.5 | 0 | 0 | 0 | 0 | 0 | 0 | 0 | 0 | 0 | 0 | 0.44 | 0 | 0 | 0.22 | 0 | 0 | 0 | 0 | 0 | 0.33 | 9 |
| | 4.3 | 0 | 0 | 0 | 0 | 0 | 0.18 | 0 | 0 | 0.06 | 0 | 0 | 0.06 | 0.06 | 0 | 0 | 0 | 0.12 | 0 | 0 | 0.41 | 17 |
| | 4.4 | 0 | 0 | 0 | 0 | 0 | 0 | 0 | 0 | 0 | 0 | 0 | 0 | 0.5 | 0.5 | 0 | 0 | 0 | 0 | 0 | 0 | 2 |
| | 5.1 | 0 | 0 | 0 | 0 | 0.04 | 0.02 | 0 | 0 | 0 | 0 | 0.45 | 0.04 | 0.05 | 0.02 | 0 | 0 | 0 | 0 | 0.02 | 0.38 | 56 |
| | 5.3 | 0 | 0 | 0 | 0 | 0.05 | 0.4 | 0 | 0 | 0.05 | 0 | 0.05 | 0.05 | 0.15 | 0.05 | 0 | 0 | 0 | 0 | 0 | 0.2 | 20 |
| | 5.4 | 0 | 0 | 0 | 0 | 0.02 | 0.06 | 0 | 0 | 0 | 0 | 0.06 | 0 | 0.61 | 0.02 | 0 | 0 | 0 | 0 | 0 | 0.22 | 49 |
| | 5.6 | 0 | 0 | 0 | 0 | 0.03 | 0.14 | 0 | 0 | 0 | 0 | 0.24 | 0 | 0.1 | 0.17 | 0 | 0 | 0.03 | 0 | 0.03 | 0.24 | 29 |
| | 5.7 | 0 | 0 | 0 | 0 | 0 | 0.2 | 0 | 0 | 0 | 0 | 0 | 0 | 0 | 0 | 0.3 | 0.1 | 0 | 0 | 0 | 0.4 | 10 |
| | 8.1 | 0 | 0 | 0 | 0 | 0.17 | 0 | 0 | 0 | 0 | 0 | 0 | 0 | 0.07 | 0 | 0 | 0.57 | 0.03 | 0 | 0 | 0.17 | 30 |
| | 8.2 | 0 | 0 | 0 | 0.04 | 0.13 | 0 | 0 | 0 | 0.04 | 0 | 0 | 0 | 0.17 | 0 | 0 | 0.04 | 0.17 | 0 | 0 | 0.39 | 23 |
| | 8.6 | 0 | 0 | 0 | 0 | 0.17 | 0 | 0 | 0 | 0 | 0 | 0.17 | 0 | 0 | 0 | 0 | 0 | 0 | 0 | 0 | 0.67 | 6 |
| | 8.7 | 0 | 0 | 0 | 0 | 0 | 0.03 | 0 | 0 | 0 | 0 | 0.03 | 0 | 0 | 0 | 0 | 0 | 0 | 0 | 0.94 | 0 | 34 |
| | NC | 0 | 0 | 0 | 0 | 0.04 | 0.03 | 0 | 0 | 0 | 0 | 0.03 | 0 | 0.05 | 0 | 0 | 0.01 | 0.01 | 0 | 0.02 | 0.79 | 450 |
| | | 0 | 0 | 0 | 0.05 | 3.23 | 2.65 | 0 | 0 | 0.27 | 0 | 1.86 | 0.19 | 2.11 | 1.04 | 0.3 | 1.03 | 0.37 | 0 | 1.02 | 5.91 | |

**Figure 4.** The confusion matrix for experiment E1—classification at individual CoD level.

Experiment E2 was conducted to assess the performance of CoD group level classification. The experiment followed the same validation protocol as E1. Except for group 4 (with only 19 instances), all other CoD groups showed consistently good performance, with values on the main diagonal varying from 0.61 to 0.66 (Figure 5). The number of CoD instances varied from 93 for group 8 to 164 for group 5, indicating that the model is robust if provided with the proper number of training examples. Moreover, better performance than E1 indicates a distinct semantic structure describing causes in individual groups. Such behavior implicitly proves that CoD were properly grouped and that the expert labeling protocol was valid. Experiment E3 was conducted to validate if the automatic segmentation of the MoM statements per project Entity is possible, which would be used in experiments E4 and E5. A model for classifying statements per project Entity (bridge, route, tunnel, and misc.) was created using the protocol described in Section 5.2.1, and its performance is presented in a new confusion matrix (Figure 6).

| | | predicted | | | | | | | total |
|---|---|---|---|---|---|---|---|---|---|
| | | **1** | **3** | **4** | **5** | **8** | **NC** | | |
| **actual** | **1** | 0.64 | 0.02 | 0.02 | 0.06 | 0.06 | 0.2 | | 122 |
| | **3** | 0.03 | 0.61 | 0.03 | 0.24 | 0.02 | 0.07 | | 99 |
| | **4** | 0.21 | 0.05 | 0.21 | 0.16 | 0 | 0.37 | | 19 |
| | **5** | 0.04 | 0.1 | 0 | 0.64 | 0.02 | 0.19 | | 164 |
| | **8** | 0.12 | 0.03 | 0.01 | 0.05 | 0.66 | 0.13 | | 93 |
| | **NC** | 0.06 | 0.05 | 0.01 | 0.1 | 0.04 | 0.74 | | 450 |

**Figure 5.** The confusion matrix for experiment E2—classification at CoD group level.

| | | predicted | | | | | total |
|---|---|---|---|---|---|---|---|
| | | **B** | **R** | **T** | **M** | | |
| **actual** | **B** | 1 | 0.02 | 0.02 | 0 | | 134 |
| | **R** | 0 | 0.99 | 0.01 | 0 | | 2575 |
| | **T** | 0 | 0.01 | 0.98 | 0 | | 162 |
| | **M** | 0 | 0.88 | 0.12 | 0 | | 41 |

**Figure 6.** The confusion matrix for experiment E3—classification at the Entity level.

The confusion matrix shows overall excellent performance for all three main Entities. Miscellaneous Entity statements were mainly assigned to the route Entity. Such behavior was expected because the miscellaneous class had neither a distinct semantic structure nor a sufficient number of instances. The E3 results show that the automatic creation of a segmented model is possible.

5.2.4. Experimental Results for Segmented and Expanded Models (E4 and E5)

Segmented (E4) and Segmented Expanded (E5) CoD classification models were created with the protocol described in Section 5.2.1, based on the E3 results. Models E4 and E5 were executed with segmentation only for the route because the other two Entities did not have a sufficient number of instances for model training and validation.

Recall as the ratio of correctly recognized instances among all instances assigned to a particular class provides insight into the effect of creating a segmented model (Table 7). E4 shows an overall increase in the recall for the most numerous CoD and a decrease for the less represented ones. The reason for this trend is twofold: the segmentation of a particular project Entity further decreases the number of available training instances, while on the other hand, the language describing CoD becomes more homogenous. The overall model performance remains similar concerning the general model.

**Table 7.** First two columns: CoD and the number of instances. Third and fourth columns: comparison of the recall metric for experiments E1 and E4. Fifth and sixth column: comparison of F-Measure metric for experiments E4 and E5.

| Delay Cause | Instances | E1 Recall | E4 Recall | E4 F-Measure | E5 F-Measure |
|---|---|---|---|---|---|
| 1.1 | 13 | 0 | 0 | 0 | 0 |
| 1.2 | 12 | 0 | 0 | 0 | 0.1 |
| 1.3 | 12 | 0 | 0 | 0 | 0 |
| 1.4 | 9 | 0 | 0 | 0 | 0.17 |
| 1.5 | 52 | 0.69 | 0.71 | 0.51 | 0.48 |
| 3.2 | 55 | 0.7 | 0.71 | 0.56 | 0.49 |
| 3.4 | 7 | 0 | 0 | 0 | 0 |
| 3.5 | 7 | 0 | 0 | 0 | 0 |
| 4.3 | 12 | 0.06 | 0 | 0 | 0.18 |
| 4.4 | 2 | 0 | 0 | 0 | 0 |
| 5.1 | 40 | 0.45 | 0.57 | 0.42 | 0.38 |
| 5.3 | 13 | 0.05 | 0 | 0 | 0.15 |
| 5.4 | 35 | 0.55 | 0.57 | 0.4 | 0.4 |
| 5.6 | 22 | 0.17 | 0.05 | 0.07 | 0.22 |
| 5.7 | 9 | 0.3 | 0.11 | 0.18 | 0.57 |
| 8.1 | 29 | 0.57 | 0.69 | 0.67 | 0.7 |
| 8.2 | 13 | 0.17 | 0 | 0 | 0.11 |
| 8.6 | 3 | 0 | 0 | 0 | 0 |
| 8.7 | 31 | 0.94 | 0.97 | 0.88 | 0.72 |

A performance comparison between E4 and E5 shows an overall increase in F-Measure for the less represented CoD classes. Expansion of the training set through abstractive summarization increases the impact of CoD with fewer samples by diversifying its semantic structure with synonyms and words with similar meanings. Although the number of training instances is increased for all classes, the more frequent ones do not benefit from it. On the contrary, the majority displays somewhat decreased performance, probably because the number of properly represented classes increases.

E4 and E5 have a similar overall performance to the general model, but the underlying logic differs. Segmentation in E4 further strengthens more frequent CoD, while abstractive summarization in E5 does the same for the less frequent CoD. The selection of proper protocol depends on a specific project requirement—if a more robust CoD detection is needed, E4 is more suited. On the other hand, E5 would be advised if checking for as many CoD as possible is a priority.

*5.3. Temporal Distribution—Root Causes of Delay Discovery in the Project*

The temporal distribution represents a unique feature that separates this research, introducing time as a valuable new level of analysis, which enables the detection and analysis of *root causes of delay*. Tracking CoD throughout the project will be useful in finding hidden trends, as well as in identifying the root cause of a delay. Some authors suggest tracking the risks concerning the location and time of occurrence and adopting an accurate risk response strategy [54]. Moreover, a study [42] emphasizes the importance of tracking key phrases through time to extract relevant information related to the observed concept. CoD-related results obtained by DREAM can be tracked and analyzed similarly, which is discussed in this section.

Due to MoM nature, it is possible to isolate and track CoD occurrences through time. CoD tracking throughout the project duration provides more information about delays in the project, which is often not the subject of official claims.

Figure 7 shows the temporal distribution of the top six most frequent CoD for the *route* Entity. The chart illustrates CoD representation in the project duration, based on its frequency in MoM statements organized by quarters. The following trends can be observed:

- CoD *rework due to errors or poor quality during construction* (5.1) is present throughout the project. The continuous representation of this cause in MoM may indicate the low performance of the contractor. Furthermore, three out of the top six most frequent CoD (3.2 and 5.6) relate to the contractor, which supports the claim of poor contractor performance. Moreover, such results may indicate the low performance of subcontractors and poor coordination by the contractor on the construction site.
- *CoD delays in the preparation or modification of design documentation during construction* (1.5) is present in most projects and can be linked to the low quality of design. In relation to (5.1), incremental trend changes may indicate the importance of this CoD for the entire project. The representation trend can be linked to the performance of the design team. Higher positive and negative steepness may indicate a high "degree of resolution" and agility of the design team during the execution phase. Based on the temporal distribution, it is also possible to analyze the performance of the design team, which is difficult to achieve with a survey.

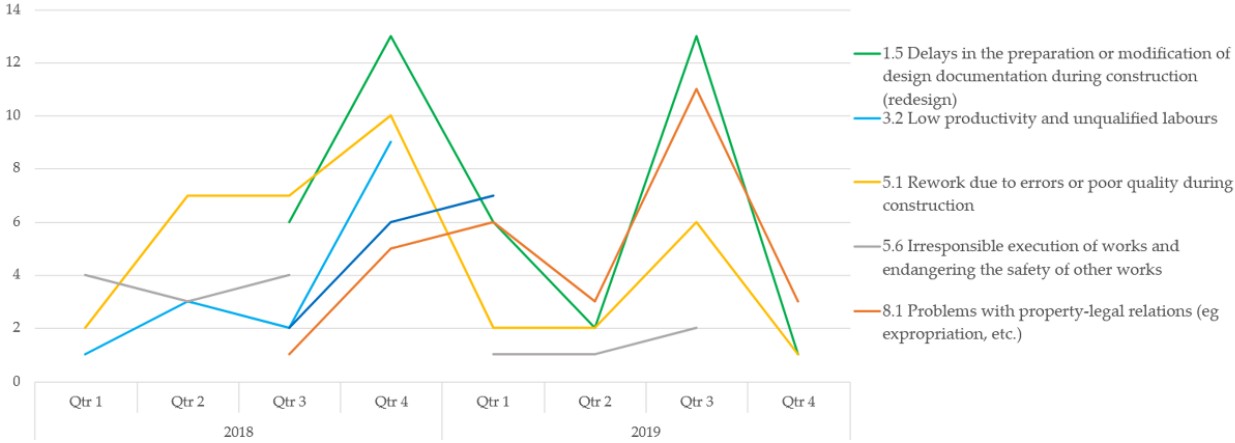

**Figure 7.** Temporal distribution of frequencies for the six most frequent CoD.

The chart in Figure 7 is based on the absolute values of CoD occurrences in MoM. On the other hand, relative CoD representation can provide additional information. Paul Parsons highlights the importance of adaptive dynamic views for information with a temporal nature to overcome the cognitive bias acquired by a static data presentation [55]. The histogram in Figure 8 shows a relative CoD representation that is also for MoM organized by quarters.

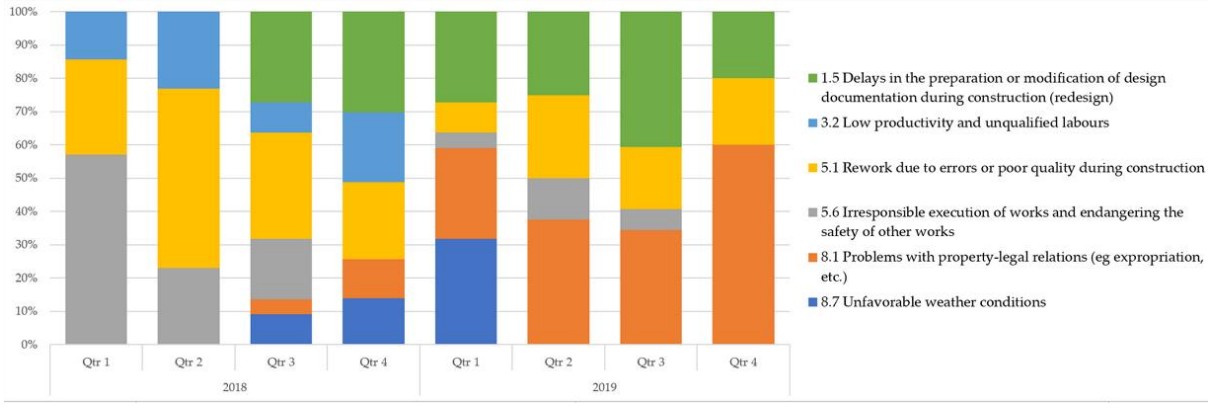

**Figure 8.** Relative temporal distribution for the six most frequent CoD—values represent relative ratios of CoD-related statements in corresponding quarters.

The CoD *problems with property–legal relations* (8.1) increases its relative representation as the project execution progresses, even though its absolute representation decreases. The analysis based on absolute values (Figure 7) shows that this CoD has a trend of reduced representation. This conclusion does not have to be solely a consequence of the actual CoD representation but can also relate to the reduced number of MoM as the project nears the end. However, the relative representation indicates that the problems with property–legal relations grow as the project progresses and reaches their maximum by the end of the project. This trend suggests a high probability of CoD 8.1 contributing to the project delay, which happened in the case project.

The relative display can help overcome the limitations originating from the frequency of meetings throughout the project, which can affect the results in some cases. In addition, observing the results in a relative environment can be a basis for discussing the mutual causality for different CoD. However, this will be a subject of future research.

Temporal distribution charts show how CoD are developing over time. Chart trend analysis enables experts to understand CoD nature, defined by their duration, intensity, and interrelations, allowing them to determine *what exactly went wrong* and to discover the root causes of delay.

## 6. Discussion

This section aims to review and compare the benefits and limitations of the existing approach and DREAM, focusing on the single project-level application. The capabilities of both are shown in Table 8.

**Table 8.** A comparative analysis of the two approaches covered by this research.

| | Existing Approach—Survey (Expert Knowledge) | Proposed Approach—DREAM (Machine Learning and Expert Knowledge) |
|---|---|---|
| Single project level (in general) | ✓ (with limitations) | ✓ |
| Single project level (in detail): | | |
|     CoD list | ✓ (with significant bias) | ✓ (with bias reduced to labeling) |
|     Project entities | ✓ (with difficulty) | ✓ |
|     Temporal distribution | X | ✓ |
|     Root CoD | ✓ (with difficulty and bias) | ✓ |

1.  Generally speaking, the existing approach can be applied at the single project level but with certain limitations. The validity of the survey would be debatable due to the small sample size because there are only a limited number of available experts familiar with the project who could participate in the survey. Inevitably, there is inherent bias when consulting stakeholders having different roles in the project. DREAM is applicable on a single project level without such limitations.
2.  Both approaches can deliver a CoD list (see Figure 1). Surveys provide a CoD list based on expert opinion (Section 4). The Spearman rank correlation values indicate a lack of consensus among stakeholders about CoD in the case project. The assumption regarding subjectivism and bias of different project participants was correct, making it difficult to reach general conclusions about CoD at the single project level.

On the other hand, the experimental results of DREAM (E1, E2, and E3) show that MoM is a good choice of documentation for describing causes of delay. As the CoD list created by DREAM (Section 5) uses project documentation and Machine Learning, the subjectivism and expert bias are significantly reduced to labeling text only.

Although not directly comparable (Spearman rank correlation vs. Confusion matrix), the case project experimental results indicate that DREAM is more suitable for CoD analysis at the single project level. The main difference is the nature of using expert knowledge. The existing approach uses experts for conducting surveys, while DREAM uses experts for labeling project documentation. For surveys, experts had to be engaged in the actual project (and thus biased). In contrast, for impartial text labeling, they only have to be familiar

with projects in general, not with the particular project (no bias). This difference leads to reduced subjectivism and a more extensive expert base in the case of using DREAM.

3.  Conducting a segmented analysis of MoM statements per project Entity proved possible, as shown in the E3 experimental results. The assumption of different distributions of delay causes for separate entities (tunnel, route, and bridge) is correct, therefore, it is a valuable addition to the in-depth analysis of the causes of the delay and one of the contributions of DREAM. Such an analysis, if made by experts solely on opinion, would be hard to perform, uncertain, and laborious.

4.  The temporal distribution of CoD is a unique feature exclusive to DREAM. Tracking the causes of delays over time, enabled by MoM dates, can be viewed as the project's heartbeat regarding problems (see Figure 7). Informative graphs (Figures 7 and 8) offer deep insight into the nature of CoD, defined by their duration, intensity, and interrelations. Furthermore, temporal distribution is a step towards defining new measures for describing individual CoD, besides their frequency detected in lists.

Identification of the temporal distribution of CoD by experts without using Machine Learning would be theoretically possible but highly impractical—time-consuming, demanding, and therefore error-prone. On the other hand, the role of DREAM experts is to interpret results, which is a rational usage of their costly working time.

5.  Finally, enabling the detection of the root causes of delay is the ultimate goal of this research. DREAM cannot detect root causes automatically, but the CoD list combined with informative graphs (Figures 7 and 8) provides experts with enough information to reconstruct the behavior of project participants and eventually enable them to reach a reliable conclusion regarding the root causes of delay.

In the existing approach, due to inherent bias, detecting root causes of delay based only on the general CoD list is debatable.

The proposed DREAM represents the "best of two worlds"—focused expert knowledge where a machine cannot interpret visual patterns and robust and unbiased Machine Learning where an expert cannot physically examine extensive documentation.

## 7. Conclusions

Due to numerous reasons, construction projects often fail to achieve their goals. Among other problems, time overruns happen frequently. Detecting causes of delays (CoD) is the first step in eliminating or mitigating potential delays in future projects. The goal of this research is unbiased CoD detection at a single project level, with the ultimate goal to enable the discovery of root causes of delay, both recognized as significant in the extensive literature review.

Two things are required to perform experiments, compare different approaches, and validate results: a general CoD list and a case project. The CoD was formed by literature review, pilot study, and filtering, finally containing 54 CoD grouped into 8 categories, presenting a minor contribution by itself. The case project was a 65km highway project with a documented 42% delay.

The point of departure for this research was the existing method of CoD detection based on expert knowledge used in surveys. The outcome of the expert-based approach is the creation of a CoD list for projects in general or groups of similar projects (e.g., grouped by type, region, and project phase). When applied to a single project, it is burdened with bias, as shown on a case project returning low Spearman Rank correlation values. For the same reasons, detecting root causes of delay based only on the general CoD list is debatable.

This research introduces a different approach—using Machine Learning techniques to detect CoD in textual project documentation, thus reducing bias, and expert knowledge to interpret Machine-Learning-generated lists and graphs to discover the root causes of delay. The key components of the proposed DREAM model are:

*   MoM as a chosen documentation type for model training;
*   Expert labeling procedure of CoD-related text segments;

- Transformer models used for CoD detection.

Expert labeling is done once for certain project types to capture semantic specifics and to train the model, which is then used on individual projects.

In the first phase, DREAM outputs a CoD list based on occurring frequency in MoM with acceptable recall values (0.69 for most frequent), significantly reducing expert-induced subjectivism to text labeling only. One of the DREAM's contributions is conducting a segmented analysis of MoM statements per project entities (e.g., tunnel, route, and bridge for road projects) which have different distributions of delay causes, enhancing CoD in-depth analysis. The model was trained for road infrastructure projects. The model also applies to other types of projects if retrained with labeled relevant CoD text sources.

In the second phase, enabled by MoM dates, DREAM adds another dimension to delay analysis—temporal CoD distribution, a unique feature exclusive to DREAM and a valuable scientific and practical contribution. The temporal distribution of CoD frequencies can be presented graphically in absolute or relative values. Although DREAM cannot detect root causes automatically, the significance of temporal distribution is in allowing experts to analyze graphs and discover the nature of CoD, defined by their duration, intensity, and interrelations, which enables them to detect root causes, the ultimate research goal. The temporal distribution is also an opportunity to develop new categorizations of CoD.

DREAM provides synergy between expert knowledge and Machine Learning and a rational way to use experts—for text labeling and interpreting results. The ability to detect root causes of delay provides project managers with the decision-making support for efficient and focused delay prevention in future projects, proving that AI enables overcoming disadvantages from conventional approaches on manual observation which is more prone to bias and confounding [10].

After discussing the benefits and limitations of both approaches, it can be concluded that DREAM is superior to expert-based surveys in single project level application and detecting root causes of delay.

The proposed model also has certain limitations directing possible future research. The DREAM uses Machine Learning and requires experts to provide training input, thus presenting a potential risk of introducing bias and subjectivism. If the text sources used for model fine-tuning are imbalanced or irrelevant, the resulting CoD detection will have diminished performance. To avoid such limitations, a set of formal procedures will be the subject of our future research:

- Guidelines for selecting the relevant text sources and their labeling according to CoD relevance;
- Access policy for creation, removal, or modification of labeled text entries used for model training;
- Benchmarks for newly trained models compared with the best performing existing models.

**Author Contributions:** M.Z.I.—conceptualization, investigation, methodology, writing—original draft preparation; Đ.N.—software, formal analysis, writing—review and editing; Z.S.—writing—review and editing; D.M.—supervision; N.I.—supervision; N.S.—data curation and visualization. All authors have read and agreed to the published version of the manuscript.

**Funding:** This research received no external funding.

**Institutional Review Board Statement:** Not applicable.

**Informed Consent Statement:** Not applicable.

**Data Availability Statement:** Not applicable.

**Conflicts of Interest:** The authors declare no conflict of interest.

## Appendix A

**Table A1.** The General CoD List.

| | Group | Code | Cause of Delay (CoD) |
|---|---|---|---|
| 1. | Design | 1.1 | Non-compliance of the project with the environmental conditions |
| | | 1.2 | Lack of details and specifications in the design documentation |
| | | 1.3 | Designed complex or inappropriate performance technology |
| | | 1.4 | Non-compliance of parts of the design documentation |
| | | 1.5 | Delays in the preparation or modification of design documentation during construction |
| | | 1.6 | Poor bill of quantities (BoQ) |
| 2. | Procurement | 2.1 | Contract award criteria—award the project to the lowest bidder |
| | | 2.2 | Contract award criteria—duration as a parameter for bid evaluation data |
| | | 2.3 | Long period of additional contracting for unforeseen and subsequent works |
| 3. | Resources | 3.1 | Lack of laborers |
| | | 3.2 | Low productivity and unqualified laborers |
| | | 3.3 | Lack of material in market |
| | | 3.4 | Delay in material and equipment delivery |
| | | 3.5 | Inadequate quality of material |
| | | 3.6 | Equipment breakdowns and obsolete equipment |
| | | 3.7 | Lack of equipment (machine) |
| 4. | Employer | 4.1 | Delays in payment by the owner |
| | | 4.2 | Change orders by owner during construction |
| | | 4.3 | Slowness in decision-making process by owner |
| | | 4.4 | Poor communication and coordination by owner and other parties |
| | | 4.5 | Lack of finances or lengthy procedure for financing unforeseen works |
| | | 4.6 | Delay to furnish and deliver the (part) site to the contractor by the owner |
| 5. | Contractor | 5.1 | Rework due to errors or poor quality during construction |
| | | 5.2 | Poor financial condition of the contractor |
| | | 5.3 | Ineffective planning and management of project by contractor |
| | | 5.4 | Poor communication and coordination by contractor with other parties |
| | | 5.5 | Inadequate contractor experience |
| | | 5.6 | Irresponsible execution of works and endangering the safety of other works |
| | | 5.7 | Delay of subcontractor |
| 6. | Consultant | 6.1 | Poor communication by consultant with other construction parties |
| | | 6.2 | Lack of experience of consultant |
| | | 6.3 | Insufficient of consultants |
| | | 6.4 | Consultant avoids taking a proactive role and issuing instructions |
| | | 6.5 | Delays in reviewing and verifying the work performed |
| | | 6.6 | Delays in reviewing and verifying the Method Statement |
| | | 6.7 | Delays in reviewing and verifying the material |
| 7. | Project | 7.1 | Original contract duration is too short |
| | | 7.2 | Inadequate or imprecise contract conditions |
| | | 7.3 | Unresolved claims, variations and VEP |
| | | 7.4 | High complexity of the project |
| | | 7.5 | Legal disputes between various parts during construction |
| | | 7.6 | Inadequate cash flow |
| | | 7.7 | Poor contract management of project |
| | | 7.8 | Lack of risk management |
| | | 7.9 | Accidents during construction |
| 8. | External | 8.1 | Problems with property–legal relations (e.g., expropriation, etc.) |
| | | 8.2 | Delay in obtaining permits and approvals from the competent authorities |
| | | 8.3 | Changes in government regulations and laws |
| | | 8.4 | Corruption and unstable political situation in the country |
| | | 8.5 | Exchange rate fluctuation (price fluctuations, cost escalation) |
| | | 8.6 | New environmental restrictions or unforeseen circumstances |
| | | 8.7 | Unfavorable weather conditions |

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
