# Peer review of "Detection and In-Depth Analysis of Causes of Delay in Construction Projects: Synergy between Machine Learning and Expert Knowledge"

_sustainability, doi:10.3390/su142214927_

Round 1

Reviewer 1 Report

The topic is of potential interest for readings. However, the presentation of results is not described with the sufficient detail. The suggested corrections are listed in the next paragraphs.

[1]     All the figures in the manuscript are not clear.

[2]     Spearman Rank Correlation non parametric test is a key method of this investigation. The author needs to describe this method in detail for the convenience of readers.

[3]     In section 2.2: Overall the text mining part remains too detailed. It is more appropriate to make it more general, referring to the main results obtained in the field, but without reporting the names of the various authors mentioned in the references.

[4]     In section 4.2: The obtained results of questionnaire should be elaborated in detail.

[5]     In section 5.1.2: The role of expert labels should be elaborated in detail.

[6]     In Section 6: The author mentioned "For surveys, experts had to be engaged in the actual project……, not with the particular project(no bias)." I was wondering if the obtained results are correct? Will the results be biased if the experts don't know the specifics of a particular project?    

[7]     Typos must be corrected: repetition of subheading 4.2; Table 9 is not found in the manuscript.

Author Response

The authors are grateful to the Reviewer for the time dedicated to revising our paper and the provided positive feedback and comments. We have done our best to implement all suggested changes to the manuscript and we are certain this has helped improve its quality. Our responses to the review comments are below, and changes that we have made to the paper are highlighted in the revised manuscript submission.

Reviewer 2 Report

- Inappropriate usage of "chapter", please revise it.

- In section 4.2, line 280, you mention that  "(2) General CoD List generated in the previous chapter". However, there is no list in the previous section, thereby, should mention in Appendix A.

- Please justify the adoption of the type Four-point Likert scale, why not other types? Justify and provide a methodological reference.

- Please replace the citation (No 19) with an appropriate reference for the statement in line number 288 (e. g., methodological reference)

- Section 5 and Subsection 5.1.2, please put them in an appropriate position (e. g., move it to the next page).

- The contents inside the figures are not clear (unreadable), please clarify them.

- For the pilot study, what did you exist from it or what are the changes? Please specify.

Author Response

(The authors gave the same response as above.)

Reviewer 3 Report

The paper presents use of Machine Learning for detection of Causes of Delay, which falls under Digitization theme of this special issue, and hence is an acceptable topic. The manuscript itself is decently drafted. I find the paper novel with an average contribution to the field. No major changes are required. However, the authors are suggested to go through the paper once again for improvement in flow and sentence structures. 

Author Response

The authors are grateful to the Reviewer for the time dedicated to revising our paper and the provided positive feedback. We have done our best to implement all suggested comments to the manuscript and we are certain this has helped improve its quality. The manuscript was proof-read by a native English language editor and the text of the manuscript has been improved.

Round 2

Reviewer 1 Report

Accept in current form.